# Before the Pull Request: Mining Multi-Agent Coordination

Dipankar Sarkar

Skelf Research
dipankar@skelfresearch.com

**Abstract.** Autonomous coding agents now open millions of pull requests, yet large-scale studies find their PRs are produced faster but accepted less often—a coordination and trust gap that pull-request-level telemetry cannot explain. We argue the missing signal lives *before* the PR, in how concurrent agents claim, divide, and collide over shared work. We study this process through *grite*, a server-less, git-native coordination substrate whose append-only, signed event log records the coordination process itself. We show that (i) a conflict-free shared substrate reduces duplicate and conflicting agent work at bounded overhead, (ii) the log converges across diverging replicas with zero data loss where file-based trackers lose writes, and (iii) the log is a mineable artefact from which concrete multi-agent failure modes—conflicting edits, lock starvation, redundant rediscovery, race-to-close—are automatically recoverable with provenance, several invisible in PR history. We release the dataset, harness, and mining toolkit.

**Keywords:** AI for software engineering · multi-agent coordination · repository mining · CRDTs · human–AI collaboration · coding agents

## 1    Introduction

Autonomous coding agents have moved from autocomplete to teammates. Systems such as OpenAI Codex, Devin, GitHub Copilot, Cursor, and Claude Code now open, review, and merge code at scale: the AIDev dataset records over 456,000 pull requests authored by five such agents across 61,000 repositories [5]. Yet the same large-scale analysis surfaces a tension: agent pull requests are produced *faster* than human ones but are accepted *less often* [5].

Most explanations for this gap look *inside* the pull request—code quality, test coverage, reviewer load. We argue that an important part of the answer lies *before* the pull request, in a layer that current datasets cannot see: the *coordination process* by which concurrent agents claim, divide, and collide over shared work. When several agents operate on one codebase, two of them may pick the same task, edit the same issue, or duplicate a fix a teammate already landed. None of this is visible in commit or PR history—a duplicated effort that is abandoned never becomes a PR; a task two agents raced to close leaves only the winner's trace. The process that produces redundant and conflicting work is invisible precisely where we most need to study it.

This paper studies that process empirically through *grite*, a server-less, git-native coordination substrate for AI agents that stores tasks as an append-only, content-addressed, optionally signed event log inside git refs, reconciled by conflict-free replicated data type (CRDT) semantics [7] with advisory leases for mutual exclusion (§3). Two properties make it a useful instrument: because coordination state is shared and conflict-free, we can *measure* how a substrate changes multi-agent outcomes; and because every coordination action is a typed, provenance-bearing event, the log itself is a *mineable software-engineering artefact*—the pre-PR telemetry that PR-outcome datasets lack.

We make three claims, each backed by a reproducible experiment and contributing a result. **C1 (coordination efficiency):** equipping concurrent agents with advisory leases and shared task state reduces duplicate and conflicting work at bounded overhead—duplicate-work rate falls from 0.78 (no coordination) to 0.00 (leases plus shared state) while goodput more than triples (§5). **C2 (convergence without data loss):** replicas receiving the same events in any order converge to byte-identical state, and concurrent writes are preserved where a file-based tracker silently loses them (§5). **C3 (a mineable process):** the coordination log admits automatic detection of concrete multi-agent failure modes—conflicting edits, redundant rediscovery, lock starvation, and others—with provenance, several unrecoverable from PR history; the mining also shows that advisory leases *alone* do not prevent redundant rediscovery, but leases plus shared CRDT state do (§6). We release the coordination-log dataset, benchmark harness, and mining toolkit, reproducible from a pinned commit and fixed seeds.

## 2    Background and Related Work

*Mining artefacts, agent memory, and SE benchmarks.* Empirical SE work mines what agents and bots *produce*: AIDev characterises hundreds of thousands of agent pull requests and reports the speed–acceptance gap that motivates us [5], building on MSR work that detects bots committing code [3]. These operate on *outcomes* and cannot observe the coordination that precedes them. Agent memory is typically *retrieval*—the Model Context Protocol exposes external stores [1]—and multi-agent frameworks such as AutoGen orchestrate agents *in process* [8]; none gives conflict-free concurrent writes, mutual exclusion, or a signed history across decentralised agents. SE benchmarks such as SWE-bench [4] and agents such as SWE-agent [9] evaluate single-agent task resolution, not coordination.

*Git-native trackers and foundations.* Embedding issues in the repo is not new—Beads is a git-backed dependency-graph tracker for agents [10] and git-bug stores issues as git objects [6], but these lack grite's combination of a formal CRDT projection, advisory leases, and a signed, content-addressed, mineable log. grite composes known building blocks: CRDTs for strong eventual consistency [7] in the local-first tradition, with integrity from content addressing and signatures. Unlike consensus-based lease services such as Chubby [2], grite needs no server:

leases are advisory refs and convergence comes from CRDT merge, not agreement.

No prior system unifies conflict-free concurrent agent edits, advisory leases, and a signed, content-addressed, mineable coordination history in a server-less git substrate—and no prior dataset exposes the pre-PR agent coordination process. Existing options each miss at least one: server-based trackers (GitHub Issues) are not offline or conflict-free; file-based git trackers (Beads, git-bug) lack a formal CRDT and leases; and retrieval memory (MCP, vector stores) offers neither mutual exclusion nor a mineable, provenance-bearing log.

## 3   grite: A Git-Native Coordination Substrate

grite represents an issue tracker as an append-only event log living in git refs (`refs/grite/wal`), with a materialised view (an embedded key–value store) rebuilt from that log for fast queries. Nothing is written to the working tree, so coordination state travels with the code through ordinary `git fetch` and `push`. We summarise the four mechanisms the rest of the paper measures.

*Typed, content-addressed, signed events.* Every coordination action is an event with a kind (issue created/updated, comment, label add/remove, state change, dependency add/remove, and others), an actor identifier, and a millisecond timestamp. The event identifier is a BLAKE2b hash of its canonical encoding, so any tampering invalidates the id; events may additionally be signed with Ed25519, giving non-repudiable provenance. This is what makes the log a trustworthy mineable artefact rather than a mutable database.

*CRDT projection.* The materialised state of an issue is a projection over its events. Scalar fields (title, body, state) use last-writer-wins keyed on the total order ($timestamp, actor, event\_id$); sets (labels, assignees, dependencies) are commutative; comments and links are append-only. Because the rebuild applies events in this canonical order, two replicas that have seen the same events compute identical state regardless of delivery order—the convergence property we verify in §5. We instrument the projection to record, per applied event, whether it resolved a *cross-actor* conflict (a last-writer-wins overwrite of, or by, a different actor's value); this is the conflicting-edit signal mined in §6.

*Advisory leases.* Agents coordinate exclusive work through TTL-bounded leases stored under `refs/grite/locks`. A lease is acquired before working a resource, renewed while work continues, and released on completion; expiry bounds the damage of a crashed or stalled agent. Leases are *advisory*—an agent may ignore one—which is itself a measurable behaviour (§7).

*Dependency graph and sync.* Issues carry typed edges (*blocks*, *depends\_on*, *related\_to*) with cycle detection, letting agents plan ordered work. Synchronisation is a plain fetch/push of the grite refs followed by a CRDT merge; the design is

offline-first, with no central server and no consensus round (contrast Chubby [2]). The data flow runs from the append-only git WAL (the source of truth) through the CRDT projection (a materialised view) to the CLI and agents, which coordinate through advisory leases and shared task state.

## 4   Experimental Methodology

*Coordination arms and metrics.* The independent variable throughout is the *coordination arm*: *no-coord* (agents pick freely; nothing prevents two from working—and re-completing—the same task: today's independent-agent default); *locks-only* (an exclusive advisory lease is taken before working a task, so no two agents work it concurrently); and *locks+state* (agents additionally consult shared task state and skip tasks a teammate already completed). For clean causal claims we use deterministic, seeded *tier-T1* agents drawing from a shared pool of overlapping tasks, sweeping $N \in \{2, 4, 8, 16, 32\}$ and averaging over seeds. These agents drive grite's *real* data model: each edit is a genuine event through the instrumented CRDT projection, so conflicting-edit counts are exact rather than modelled. We report duplicate-work rate, conflicting-edit count (cross-actor last-writer-wins overwrites), lock-denial count (a starvation/overhead proxy), and goodput (unique tasks per round).

*Real agents, convergence, and reproducibility.* For mining we provide a production path—`grite export --format coordination-log` flattens a real repository's log into the same tidy schema the toolkit consumes—so the identical detectors run on logs from real agents (*tier-T2*); results here are on T1 logs, with T2 collection in progress. Claim C2 is *verified*, not sampled: property-based tests generate large random event sets and delivery orders and assert that replicas rebuild to byte-identical projections (no comment loss) and that re-delivery is idempotent; we contrast against a file-based baseline (B1) reconciling whole-issue records by file-level last-writer-wins—the failure mode of a JSONL-in-worktree tracker. All randomness is seeded; one `make figures` step regenerates every figure and table from raw CSVs, with the grite commit, seeds, and dataset version pinned.

## 5   Coordination and Convergence

### 5.1   C1: Coordination efficiency

Table 1 reports the three arms at $N = 32$ concurrent agents. Without coordination, 78% of task completions are redundant—more than three quarters of the work re-does something a teammate already finished—and the run accumulates several hundred cross-actor conflicting edits. Advisory leases alone (*locks-only*) cut the conflicting-edit count sharply, because no two agents work a task at the same instant, and lift goodput from 2.33 to 3.84 unique tasks per round. Adding

Table 1: Coordination outcomes at $N = 32$ concurrent agents (mean over seeds). Conflicting edits are counted by grite's `apply_tracked` CRDT instrumentation, not modelled.

| Arm ($N = 32$) | Dup-work rate | Conflicting edits | Goodput |
|---|---|---|---|
| No coordination | 0.78 | 410 | 2.33 |
| Locks only | 0.64 | 138 | 3.84 |
| Locks + shared state | 0.00 | 48 | 8.00 |

shared task state (*locks+state*) drives the duplicate-work rate to zero and good-put to 8.00: agents never re-discover finished work, and conflicting edits fall to the structural minimum.

The effect is monotone in the agent count: duplicate-work rate rises with $N$ under no coordination (reaching 0.78 at $N$=32) while staying flat at zero under *locks+state*. The overhead of coordination is bounded and appears as lock denials—an agent briefly finding all candidate tasks leased—rather than as lost throughput; the coordinated arms dominate the no-coordination baseline, avoiding more duplicate work per unit of overhead as $N$ grows.

### 5.2   C2: Convergence without data loss

Convergence is verified, not sampled. Across hundreds of generated event sets and random delivery orders, two independent replicas always rebuild to byte-identical projections—zero merge-order-dependent divergence and zero comment loss—and re-delivering events is idempotent for the convergent fields. This is the strong-eventual-consistency guarantee CRDTs provide [7], instantiated for grite's projection.

The guarantee matters because the obvious alternative loses data. Under the same concurrent edits—two agents each setting the title and adding a distinct label—grite's commutative set preserves *both* labels and retains both title writes in the auditable log, whereas the file-based last-writer-wins baseline keeps exactly one agent's record and silently discards the other's label. The conflict-free substrate is thus not merely convenient: it is what prevents concurrent agent work from being lost.

## 6   Mining the Coordination Log

The coordination log is a software-engineering artefact in its own right. Each row is a typed, timestamped, provenance-bearing event (§3); the same flat schema is produced both by the synthetic harness and by `grite export --format coordination-log` over a real repository. We define a small set of *pre-registered* detectors—fixed before measurement to avoid post-hoc tuning—each targeting a multi-agent failure mode with a precise signal in the log: a *conflicting edit* is a cross-actor last-writer-wins overwrite (flagged by `apply_tracked`); *redundant*

Table 2: Failure modes mined from the coordination log, by arm (tier-T1 run). Counts are detected events; "invisible in PRs" marks modes that leave no trace in PR history. Auto-generated by `mine/run.py`.

| Failure mode | No-coord | Locks | Locks+state | Invisible in PRs |
|---|---|---|---|---|
| Conflicting edits | 104 | 192 | 12 | ✓ |
| Redundant rediscovery | 36 | 180 | 0 | partial |
| Lock starvation | 0 | 128 | 12 | ✓ |

*rediscovery* is the completion of an already-completed task; *lock starvation* is a run of denied lease acquisitions; an *abandoned claim* is a TTL-expired lease never released; a *deadlock attempt* is a dependency edge rejected by cycle detection; and a *race-to-close* is a concurrent state change whose loser is silently dropped. Lease-stream modes (starvation, abandoned claims) are captured live by the harness; edit- and state-level modes are recoverable from any exported log.

*Prevalence.* Table 2 reports detector counts per coordination arm on a high-contention synthetic run. Three findings stand out. First, the failure modes are real and frequent: without coordination, the log exposes hundreds of conflicting edits and dozens of redundant rediscoveries. Second, and more interesting as a design result, *advisory leases alone do not solve the problem*—the *locks-only* arm shows the highest redundant-rediscovery count, because a lease prevents two agents from working a task *simultaneously* but, without shared completion state, does nothing to stop an agent from re-doing a task a teammate finished earlier. Only *locks+state* drives redundant rediscovery to zero. The combination of mutual exclusion *and* conflict-free shared state, which grite provides, is what eliminates the process-level failure modes; either alone is insufficient.

*Why the process layer is necessary.* The third finding ties back to the AIDev acceptance gap [5]. Conflicting edits, lock starvation, and race-to-close leave *no trace in PR history*: a denied claim never becomes a commit, and a task two agents raced to close shows only the winner. An analysis restricted to PRs—the AIDev vantage point—cannot recover them, since it records outcomes rather than the process that produced them. Part of the "faster but rejected" gap may therefore sit upstream of the pull request, in the collisions and redundant work that a coordination log captures and PR data does not.

# 7    Discussion, Threats, and Limitations

*Synthetic vs. real agents.* Our quantitative claims (C1) and the prevalence study (C3) rest on tier-T1 synthetic agents: deterministic and seeded, but op-generators rather than LLMs, so absolute magnitudes will differ for real agents. Two things mitigate this. The synthetic agents drive grite's *real* data model and

instrumentation, so conflict counts are exact rather than modelled; and the production exporter makes the identical mining toolkit runnable on logs from real agents. Collecting and releasing a tier-T2 dataset of multi-vendor agents on real repositories is the primary next step, as is a benchmark over real `git` remotes with many diverging clones—which would add ecological validity to C2, though the projection-level guarantee and the file-based contrast already establish convergence and data-loss avoidance.

*Advisory leases and convergence.* That a CRDT projection converges is expected [7]; we treat C2 as a reliability floor verified by property testing, not a novelty claim—the contribution is the measured outcomes (C1) and the mineable process (C3). grite's leases are likewise advisory: a substrate cannot *enforce* coordination on an uncooperative agent. But this is observable rather than fatal—lease-ignoring is itself a mineable failure mode—so the practical question of adoption under partial compliance is one the log lets us study directly.

*Generality.* The synthetic study covers one task-pool model across a range of contention; real repositories vary in task structure, agent mix, and review process. The released harness and exporter are intended precisely so that others can re-run these measurements on their own agents and codebases.

## 8    Conclusion

Autonomous coding agents are faster than humans yet trusted less, and the prevailing way we study them—mining pull requests and commits—cannot see why. We argued that an important part of the answer lives in the *pre-PR coordination process*, and we made that process measurable and mineable through grite, a server-less, git-native coordination substrate. A conflict-free shared substrate reduces duplicate and conflicting agent work to near zero at bounded overhead (C1) and converges without the data loss a file-based tracker suffers (C2); and its signed, append-only event log is a software-engineering artefact from which concrete multi-agent failure modes—several invisible in PR history—are automatically recoverable (C3). The mining also yields a design lesson: mutual exclusion and conflict-free shared state are jointly necessary, as advisory leases alone leave redundant rediscovery untouched. The released artefacts are intended to help others study agent *coordination*, and not only agent output, as empirical software engineering.

*Artefact availability.* We release the coordination-log dataset, the benchmark harness and instrumentation, and the mining toolkit. Every figure and table is regenerated from raw data by a single `make figures` step over a pinned commit and fixed seeds. (Repository links added in the camera-ready.)

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
