# OpenReview forum: "Before the Pull Request: Mining Multi-Agent Coordination"
_KI/2026/Workshop/AI4SE — Submitted to AI4SE Workshop_

### Official Review · Reviewer_WCHh · 2026-06-08
**See below.**

**Rating:** 3
**Confidence:** 2

**Review:**

This paper presents grite, a lightweight Git-based system that helps multiple coding agents coordinate their work before they create pull requests, reducing duplicated or conflicting changes and making coordination problems easier to detect and analyze.

Points in favor:
- Highly relevant topic.
- Makes untransparent processes measurable.

Points against:
- Description of the experiments performed must be improved. It its current form, the paper is hard to follow.
- Discussion of threats to validity should be expanded based on Wohlin et al. (see below).


Further remarks regarding sections:

Abstract
- Results should be explained a bit more for non-experts in the field. Otherwise, it could be hard to value the contributions properly.

Introduction
- It should be better explained what a “a server-less, gitnative coordination substrate” is.
- Replace § with word section.
- From reading the paper goal, one could understand that grite is just a means to investigate some aspects of AI agents. When reading the claims, it seems to be a solution to several problems in itself. This should be made clearer.

Background and Related Work
- General: A lot of sentences are extremely long and over-complicated in terms of nesting, which makes it difficult to understand.
- Section talks about grite`s benefits without explaining what grite is in more detail beforehand.
- Wording describing the benefits to grite is a bit too selling-oriented for a research paper.
- Related Work should be discussed in more depth.

grite: A Git-Native Coordination Substrate
- Section would benefit from example data which is measured by the four mechanisms.
- It stays unclear from which research goal the measures were derived from. They can partially be mapped to the claims mentioned in Section 1, but not that clear to me.
- Paper makes a lot of forward references to later Sections, which again makes it difficult to follow.

Experimental Methodology
- The first sentence in the first paragraph is hard to follow: it goes across 6 lines and reads more like a list of things concatenated.
- Method stays somewhat unclear: What experiments were conducted using which agents on which example code?

Results
- Sections 5 und 6 should be merged into one section presenting the results of experiments performed.
- A separate section should then discuss / interpret the results in the context of the three claims made at the beginning in more detail.

Discussion, Threats, and Limitations
- Focuses on discussion of threats, which is OK.
- Threats to validity should address: Conclusion, Internal, Construct and External Validity (see Wohlin et al., Experimentation in Software Engineering)

Conclusion
- What is the planned future work?

---

### Official Review · Reviewer_Mnfr · 2026-06-12
**Unclear contribution, confusing experimental methodology**

**Rating:** 4
**Confidence:** 4

**Review:**

The paper uses a tool called grite to create coordination between agents and to produce traces before pull requests. The motivation from the abstract and introduction is to better understand what is happening before a PR was submitted to understand why so many are rejected. Throughout the paper, however, the authors look more towards what effects of the coordination in grite are.

Overall, I think there is an interesting contribution somewhere in the paper, but I found it quite confusing. As already hinted at is the storyline of the paper not clear. What problem are you addressing and what is the contribution of the paper? What is the role of grite here? You do not provide a link to the repo. Looking for it, however, I found that the author of the paper seems to be a main contributor to the project. So, is grite a contribution of this paper? If so, I would talk about it more.

The other main problem is the methodology. You mention the independent variable. That is good. But what is the data you use for the analysis? You talk about synthetic agents, but what are they doing? How do you calculate the dependent variables?